# The Perioperative Surgical Home in Pediatrics: Improve Patient Outcomes, Decrease Cancellations, Improve HealthCare Spending and Allocation of Resources during the COVID-19 Pandemic

**DOI:** 10.3390/healthcare8030258

**Published:** 2020-08-07

**Authors:** Aysha Hasan, Remy Zimmerman, Kelly Gillock, Richard H Parrish

**Affiliations:** 1Section of Anesthesiology, St. Christopher’s Hospital for Children, Philadelphia, PA 19134, USA; 2College of Medicine, Drexel University, Philadelphia, PA 19129, USA; rjz34@drexel.edu (R.Z.); kelly.jean.gillock@drexel.edu (K.G.); 3Tower Health, Reading, PA 19612, USA; parrish_rh@mercer.edu; 4Department of Pharmacy Services, St. Christopher’s Hospital for Children, Philadelphia, PA 19134, USA; 5School of Medicine, Mercer University, Macon, GA 31207, USA

**Keywords:** anesthesiology, COVID-19, nursing, pediatrics, pharmacy, surgeon, surgical home, perioperative, team-based care

## Abstract

Cancellations or delays in surgical care for pediatric patients that present to the operating room create a great obstacle for both the physician and the patient. Perioperative outpatient management begins prior to the patient entering the hospital for the day of surgery, and many organizations practice using the perioperative surgical home (PSH), incorporating enhanced recovery concepts. This paper describes changes in standard operating procedures caused by the COVID-19 pandemic, and proposes the expansion of PSH, as a means of improving perioperative quality of care in pediatric populations.

Cancellations or delays in surgical care for pediatric patients that present to the operating room create a great obstacle for both the physician and the patient. Currently, in the United States (US), an estimated 343,670 elective operations have been cancelled weekly due to COVID-19 related concerns [1]. While this number focuses on adult cases, the fact remains that a majority of elective surgeries were affected by COVID-19, alluding to an unprecedented increase in cancellations and/or delays of both adult and pediatric surgeries. The medical dollars, hours and personnel wasted in delaying or canceling a case can be avoided by instituting the perioperative surgical home (PSH), especially during the new emergence of COVID-19, and with other chronic illnesses on the rise (i.e., obesity, diabetes) [2].

The PSH model (Figure 1) is a continuous patient-centered approach that involves a multidisciplinary team of physicians and healthcare providers, aimed at individualized attention that begins when the decision for operative care is made and ends approximately 30 days after hospital discharge [3]. Not only does this increase coordination between physicians, but it provides a standardized, evidence-based approach to patient care [4]. This streamlined model decreases unnecessary testing and cancellations, providing greater operative room access for in-patients, and contributing to decreased healthcare costs. PSH in a pediatric setting allows the child to be holistically evaluated by the perioperative team, including the pediatrician. Furthermore, anesthesiologists are well equipped to lead this team, as they play a critical role in preoperative, intraoperative and postoperative care, and are essential to postoperative pain management [5].

Despite being introduced recently with its first proposal in 2011, PSH has gained traction for its impact in the United States healthcare system, but the concept of perioperative medicine exists globally [7]. France, among other European countries, adopted Enhanced Recovery After Surgery (ERAS ^®^) guidelines, to implement structured protocols with the goal of improved patient outcomes; although, standardizations vary in regard to the applied specialty. In contrast, PSH focuses on patient-centered care, lowering healthcare costs, and improving patient experience via quality of care and patient satisfaction [8]. However, due to a plethora of factors, such as increasing healthcare costs, decreasing quality of patient care, and government incentives, both France and the United States are opting toward PSH, an individualized approach to perioperative medicine [3].

Incorporating pediatric-appropriate enhanced recovery concepts, perioperative outpatient management begins prior to the patient entering the hospital for the day of surgery [8]. Many health systems and surgical organizations practice using the PSH, in which the patient is seen in a team-based approach to address any issues that may delay or prohibit surgery. Pediatric PSH has not only been known to help reduce healthcare costs and increase patient and family satisfaction, but it also lowers school absences, reduces the impact on parents and patients, and decreases hospitalizations, emergency room visits and patient length of stay [9,10]. Kash et al. further validated this claim in a study analyzing US perioperative initiatives, including PSH models and PSH-like models, where 82% of those studied were reported to have significant positive results for cost and efficiency of hospital resources, as well as clinical outcomes [11]. While this study examines adult clinical care facilities, subsequent positive responses have been observed in pediatric settings; however, a scarcity of multicentered studies exist, due to fewer instances of implementation in pediatric hospitals.

The patient’s perioperative period continues, as the patient optimizes their healthcare status with procedural education, medications, laboratory work, overcoming anxiety and inquire further with regards to their procedure. The patient enters the hospital fasting from the night before, and continues the process through admissions and entering the same day holding area. Similarly, inpatients begin this process typically the day they are admitted until they are down in the operating room. Statistically, only 4% of elective procedures were cancelled after seeing an anesthesiologist for preoperative evaluation, as compared to 11% of the patients being unseen [12].

In the current climate, we are facing, as healthcare practitioners, perioperative practice has changed drastically. Currently, many patients are required to be tested for pathogens such as coronavirus prior to entry into the operating room [13]. Patient access to primary care physicians is limited by volume, panic, fear and anxiety, exacerbated by mandated self-isolation and quarantine protocol [14]. Thus, by implementing PSH in pediatric hospitals, primary care physicians are incorporated within the perioperative team, allowing remote access to care. Patient fear and panic can be limited or avoided, while still providing adequate care.

In addition, patients and guardians are fearful and anxiety ridden, of not only the procedure they are undergoing, but also of the risk of acquiring an infection or other acute illness. Patients and their families are hesitant to admit to potential exposure of COVID-19 (large gatherings, unsafe hygiene, not wearing a mask in public, etc.) on the day of surgery for the fear of cancellation, thereby potentially increasing risk factors to providers and to themselves. The outcomes of a patient with COVID-19 receiving a surgical procedure with or without anesthesia are unknown, while in other upper respiratory infections, post-operative complications can be dire [15]. Patients may even avoid coming to the hospital altogether and avoid necessary procedures because of exposure concerns. Elective procedures are no longer performed at 100% capacity in facilities across the country, because fear dominates the healthcare sector [16]. A system conceived as onerous is designed to alleviate some of the fears, prepare the patients for surgery, test the asymptomatic patient for antibodies and high risk patients with swabs and antibody tests (in the COVID-19 era), to provide useful information to both the patient and healthcare team is already in place but underutilized. Beyond the COVID era we face today, the perioperative surgical home can capture potential infections, chronic underlying diseases that need further management, and risk assessment that can provide information to the practitioners, especially when the patient is on the operating room table. PSH is a centralized concept capable of overcoming many of these obstacles faced by the healthcare industry, especially those aimed toward the pediatric population.

PSH consists of the patient visiting the anesthesiologist in a clinic-based setting. It is comprised of helping the patient and parents understand the decision of proceeding with the proposed procedure, the risks and benefits of the procedure, and recovery afterwards. It has also helped advance the enhanced recovery after surgery initiative by beginning protocols, to enhance success in advance (i.e., adequate hydration) [17]. This model helps to identify and resolve risk stratification strategies prior to the day of surgery, allowing all teams involved from admissions, nursing, pharmacy, physicians and technical staff to be prepared for the patient, rather than hurrying to gather tools, medication and personnel the day of the procedure. It is also a vital component in pediatric hospitals. In the pediatric realm, patients are accompanied by a legal guardian that must also be involved in the patient’s care, decisions and preparation of surgery. The PSH can facilitate coordination between families and healthcare providers especially, since many individuals are often required in the care of the pediatric patient [18]. It is worth mentioning that pediatric populations may suffer from rare or congenital disorders that often require specialists or individualized care for psychosocial concerns [18,19]. In addition, treating children through the use of PSH also presents age-related challenges, due to physiological and surgical variability in comparison to adults [20]. For these reasons, it is imperative that the core focus of the perioperative period is the health of the patient rather than the physician or specialty supervising care [18]. Therefore, continuity of care must be maintained across specialties, subspecialties, and facilities.

This pandemic is a reminder that, as a hospital system, we are highly unequipped to deal with the unknown. In order to prepare for the unknown, a system in which patients are screened in advance for acute and chronic illnesses, hospitals or surgical centers is prepared for possible complications. Team members are all informed in advance, not only to improve patient healthcare, but it will also reduce wasteful costs, complications and cancellations. Many hospitals have yet to adapt and mandate the implementation of PSH. Studies show that the benefits of implementing the perioperative surgical home are cost saving [3,8,21]. With this knowledge, the perioperative team should work collaboratively to implement and mandate all patients that present for elective procedures receive clearance from the perioperative surgical home.

## Figures and Tables

**Figure 1 healthcare-08-00258-f001:**
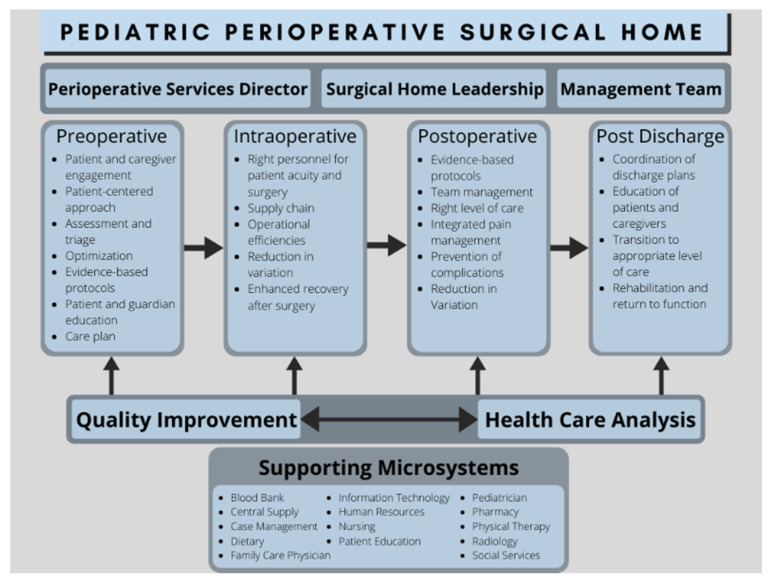
Diagram illustrating an overview of Pediatric Perioperative Surgical Home. This has been adapted for pediatric implementation from the American Society of Anesthesiologists’ Perioperative Surgical Home Collaborative [6].

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
