# Peer review of "The Perioperative Surgical Home in Pediatrics: Improve Patient Outcomes, Decrease Cancellations, Improve HealthCare Spending and Allocation of Resources during the COVID-19 Pandemic"

_healthcare, 2020, doi:10.3390/healthcare8030258_

Round 1
Reviewer 1 Report
The manuscript of Hasan et al entitled “The Perioperative Surgical Home in Pediatrics: Improve Patient Outcomes, Decrease Cancellations, Improve HealthCare Spending and Allocation of Resources during the COVID-19 Pandemic” is very interesting and very well written. Some minor points have to be improved before the acceptance.
Minor points:
- The title of the manuscript mentions pediatric patients. However, the manuscript deals with surgery in Adults and in Pediatrics. I suggest to the authors to change the title or to write a manuscript exclusively focusing on pediatric patients.
- I advise the authors to make a comparison with European countries such as France which already perform PSH or similar methods.
- A chart summarizing PSH features may help the readers to understand PSH principles
Author Response
We are very grateful for the reviewer's suggestions.
1. The title of the manuscript mentions pediatric patients. However, the manuscript deals with surgery in Adults and in Pediatrics. I suggest to the authors to change the title or to write a manuscript exclusively focusing on pediatric patients.
Response: We thank Reviewer #1 for their kind words and comments.. We amended the paper to emphasize the impact of PSH on the pediatric population. This can be seen throughout the paper with the addition of 5 additional pediatric-specific references (References 9,10,19,20,21). We modified the language of the paper to include relevant information such as involving guardians and families shown in lines 68-70, 91, 92-93, 110 as well as explicitly mentioning the pediatric population or Pediatric PSH shown in lines 12, 18, 22, 67,and 107-108. Furthermore, we made considerable changes by including pediatric specific details such as the inclusion of the pediatrician into the perioperative team in lines 51-52, the difficulties of pediatric PSH implementation in lines 120-123, and adjustment to the patient’s condition rather than dictating a specific speciality or physician must always be in charge shown in lines 123-126. Papers discussing adults were used for pediatric relevance only when no pediatric research existed and to demonstrate additional impacts the PSH can have on the pediatric population as discussed in lines 26-32 and 73-76.
2. I advise the authors to make a comparison with European countries such as France which already perform PSH or similar methods.
Response: We thank Reviewer #1 for their kind words and comments.. We discuss in lines 55-59 the connection of the PSH to France and other European countries, specifically emphasizing the Enhanced Recovery After Surgery (ERAS) guidelines. We show in lines 59-63 that while a similarity exists of an overarching perioperative protocol, PSH is much better suited to increasing quality of care due to its patient-centered care.
3. A chart summarizing PSH features may help the readers to understand PSH principles.
Response: We thank Reviewer #1 for their kind words and comments.. We have included an in-text citation in line 44 and have embedded the figure near within the text.
Reviewer 2 Report
I have reviewed the publication "The Perioperative Surgical Home in Pediatrics: Improve Patient Outcomes, Decrease Cancellations, Improve Healthcare Spending and Allocation of Resources during the COVID-19 Pandemic". I strongly feel that this publication should be accepted as this is relevant to the current crisis and the issues relating to COVID-19 pandemic.
Author Response
Responses to the comments of Reviewer #2
1. I have reviewed the publication "The Perioperative Surgical Home in Pediatrics: Improve Patient Outcomes, Decrease Cancellations, Improve Healthcare Spending and Allocation of Resources during the COVID-19 Pandemic". I strongly feel that this publication should be accepted as this is relevant to the current crisis and the issues relating to COVID-19 pandemic.
Response: We thank Reviewer #2 for their kind words and comments.
This manuscript is a resubmission of an earlier submission. The following is a list of the peer review reports and author responses from that submission.